# Testing for HIV Increases the Odds of Correct Fetal Ultrasound Result

**DOI:** 10.3390/tropicalmed7090242

**Published:** 2022-09-13

**Authors:** Carlo Bieńkowski, Małgorzata Aniszewska, Justyna D. Kowalska, Maria Pokorska-Śpiewak

**Affiliations:** 1Doctoral School, Medical University of Warsaw, Żwirki i Wigury 61, 02-091 Warsaw, Poland; 2Department of Children’s Infectious Diseases, Medical University of Warsaw, Wolska 37, 02-091 Warsaw, Poland; 3Hospital of Infectious Diseases, 01-201 Warsaw, Poland; 4Department of Adults’ Infectious Diseases, Medical University of Warsaw, Wolska 37, 02-091 Warsaw, Poland

**Keywords:** HIV testing, vertical infections, congenital infections, prevention

## Abstract

Introduction: Infectious diseases during pregnancy may pose a threat to both mother and the developing fetus. It also creates an opportunity to screen for diseases being widely underdiagnosed among women in Poland, such as human immunodeficiency virus (HIV) or sexually transmitted infections (STI). Therefore, we aimed to assess the number of pregnant women that had not been tested for HIV despite the recommendations. In addition, a comparison of clinical evaluation between HIV-tested and non-tested pregnant women was also performed. Material and methods: Medical records of all consecutive pregnant women, referred to our Infectious Diseases Hospital between September 2019 and March 2020 were retrospectively analyzed. Implementation of recommended screening testing towards infectious diseases during pregnancy including human immunodeficiency virus (HIV), hepatitis B virus (HBV), hepatitis C virus (HCV), cytomegalovirus (CMV), syphilis, and rubella, were also analyzed. Results: Medical records of 273 women were included in the analysis. The median age was 32 years (interquartile range: 26–33 years). In total 243/273 (89.0%) had been tested for HIV as recommended, and the remaining 30/273 (11.0%) had not been tested. HIV infection was not confirmed in any of the participants. Only one woman within the HIV non-tested group had been correctly tested towards other infections during her pregnancy. The recommended full testing was more likely to be correctly implemented in women who had also been tested for HIV (171/243, 70.4% vs. 1/30, 3.3%, OR 68.9; 95% CI 9.2–515.3, *p* < 0.00001). Moreover, the correct fetal ultrasound result was more likely to be obtained in women who had been tested for HIV as recommended (234/243, 96.3% vs. 11/30, 36.7%, OR 44.9; 95% CI 16.6–121.8, *p* < 0.00001). Conclusions: Despite the law regulations, 11% of pregnant women referred to consultations to the infectious diseases center had not been tested for HIV. At the same time, correct fetal ultrasound results are more likely to occur in women tested for HIV according to recommendations. This suggests that a holistic approach to screening, both for communicable and non-communicable diseases, among pregnant women may translate to better pregnancy outcomes.

## 1. Introduction

Infectious diseases during pregnancy may pose a threat to both mother and the developing fetus. Therefore, correct implementation of screening testing towards infections is crucial [1,2].

In 2018, the Ordinance of the Minister of Health on the Standard of Organizational Perinatal Care was published in Poland. Since then, every pregnant woman should undergo several laboratory tests for infectious diseases including being tested for human immunodeficiency virus (HIV) infection [3].

Poland is a low prevalence country, with a stable epidemiological situation. from the start of the study in 1985 until 31 December 2020, a total of 26 486 HIV infections were detected in Poland. Each year approximately one thousand new HIV infections are reported in Poland [4].

The aim of the study was to assess HIV testing among pregnant women who were referred to a reference center due to suspicion of toxoplasmosis. In addition, we have reviewed the characteristics of pregnant women tested and not tested for HIV in order to identify patterns related to HIV testing in pregnancy.

## 2. Materials and Methods

We performed a retrospective analysis of data collected from medical records of all consecutive pregnant women, referred by their gynecologists to the Hospital for Infectious Diseases in Warsaw due to suspected primary *Toxoplasma gondii* (TG) infection, between 1 September 2019 and 14 March 2020. In our hospital, we consult pregnant women with suspicion of toxoplasmosis and/or other infectious diseases (rarely). Therefore, in order to unify the cohort, we decided to include only women with suspected toxoplasmosis. Analyzed data included implementation of recommended screening testing for infectious diseases during pregnancy including human immunodeficiency virus (HIV), hepatitis B virus (HBV), hepatitis C virus (HCV), cytomegalovirus (CMV), syphilis, and rubella (see Table 1). Moreover, anamnesis data including age, place of residence, socioeconomic status, history of miscarriage, and comorbidities were also analyzed.

Implementation of a testing scheme for vertical infections was considered correctly performed if the first testing towards toxoplasmosis, rubella, HIV, HCV, and syphilis were performed before the tenth week of gestation. If the first tests for TG were negative, reassessment was recommended between the twenty-first and twenty-sixth week of gestation. Moreover, testing for HBV and HIV was recommended between the thirty-third and thirty-seventh week of gestation (see Table 1).

The correct fetal ultrasound result was determined when the radiologist assessed the fetus as correctly developing during the ultrasound examination.

### 2.1. Statistical Analysis

The characteristics of women tested and not tested for HIV were compared. The Mann–Whitney U test was used to compare continuous variables and the chi^2^ test with Fisher’s exact test correction where applicable was used to evaluate categorical variables, respectively. A *p*-value of <0.05 was considered significant. Statistical analyses were performed using Medcalc ver. 20.009, Ostend, Belgium. Logistic regression was used to identify factors associated with odds with correct clinical follow-up of pregnant women.

### 2.2. Ethical Statement

The design of the work conforms to standards currently applied by the Medical University of Warsaw’s Bioethics Committee. Approval number: AKBE/132/2021.

## 3. Results

Medical records of 273 women were included in the analysis. The median age was 32 years (interquartile range: 26–33 years). In 119/273 (43.6%) of the participants, the place of inhabitance was in a rural area, 44/273 (16.1%) had a history of miscarriage, and 267/273 (97.8%) had a good socioeconomic status. Chronic diseases were reported in 69/273 (25.3%) pregnant women, and 53/273 (19.4%) had autoimmune diseases. In total 243 of 273 (89.0%) women were tested for HIV as recommended. Women tested for HIV, as compared to those not tested, were not statistically different regarding: place of residence (*p* = 0.2540), socioeconomic status (*p* = 0.6530), chronic diseases (*p* = 0.1243), autoimmune diseases (*p* = 0.2231), having any symptoms during consultation (*p* = 1.0000), and history of miscarriage (*p* = 0.2546) (Table 2).

In addition, women not tested for HIV were more likely not to be tested with other recommended tests (29/30, 97.3% vs. 72/243, 29.6%, *p* < 0.00001). Only one woman not tested for HIV had been correctly tested for other infections during her pregnancy (3.3%).

The recommended testing was more likely to be correctly implemented in women who had also been tested for HIV (171/243, 70.4% vs. 1/30, 3.3%, OR 68.9; 95% CI 9.2–515.3, *p* < 0.00001). Moreover, the correct fetal ultrasound result was more likely to be obtained in women who had been tested for HIV as recommended (234/243, 96.3% vs. 11/30, 36.7%, OR 44.9; 96% CI 16.6–121.8, *p* < 0.00001) (See Table 3).

Women identified as not tested for HIV at baseline had the test performed in the course of consultation and none was positive.

## 4. Discussion

According to Drake et al. who analyzed data from 19 cohorts representing 22,803 total person-years, the pooled HIV incidence rate during pregnancy/postpartum was 3.8/100 person-years (95% CI 3.0–4.6): 4.7/100 person-years during pregnancy and 2.9/100 person-years postpartum (*p* = 0.18) [2]. In our cohort, we have not confirmed any HIV infection. However, HIV incidence among the population of pregnant women should be taken into consideration, and testing for the infection should be performed as this is the only possibility of preventing mother-to-child transmission of HIV [3,4,5]. Although testing during the first visit is recommended worldwide, reassessment during the third trimester should also be performed. Cassimatis et al. showed that despite not having the second testing recommended in their region it should be performed in order to reduce the risk of mother-to-child HIV infection [6]. Currently in Poland, there is a recommendation that provides guidelines regarding HIV testing for pregnant women [7]. Nonetheless, 11% of pregnant women in our cohort had not been tested during their first visit, despite the recommendations.

Razzaq, et al. in their systematic review on barriers to uptaking HIV testing for pregnant women concluded that in low- and middle-income countries pregnant women are facing challenges due to healthcare systems factors, lack of education, and fear of the test results [8]. These challenges may also occur in our region. Nonetheless, we did not analyze the cause of not being tested for HIV.

Women in Poland remain at high risk of late testing and diagnosis of HIV [9,10]. Prompt HIV diagnosis during pregnancy followed by appropriate intervention decreases the risk of vertical HIV transmission from over 40% to below 1% [11,12]. Therefore, implementing the recommended testing scheme for HIV infection in the first and third trimester of pregnancy is not only proposed, but required by law. As we have shown, it increases the odds of correct fetal ultrasound almost 45 times. Moreover, good lifestyle habits may prevent some other infections during pregnancy [13].

## 5. Limitations

Our cohort had been under gynecological care; nonetheless, we did not show whether these women had not been tested for HIV infection due to their own will or physician’s recommendation. In addition, it was a retrospective analysis of a cohort of women with suspected toxoplasmosis. It may not be as representative for pregnant women without suspected toxoplasmosis. Moreover, these women in order to be tested for toxoplasmosis should have been tested also for HIV according to recommendations, therefore, women who had never been tested for toxoplasmosis could not be consulted in our hospital and be included in the analysis. In addition, due to the fact that our hospital was transformed into a COVID-19 patients-only hospital in March 2020, we were unable to analyze follow-up. Therefore, our analysis is limited to 273 pregnant women.

## 6. Conclusions

Despite the law regulations, 11% of pregnant women had not been tested for HIV. The characteristic of women tested and not tested for HIV were similar and we were not able to establish a pattern associated with lower HIV test utility among patients referred to us. Correct fetal ultrasound results were more likely to be obtained in women tested for HIV according to recommendations. This suggests that a holistic approach to screening, both for communicable and non-communicable diseases, among pregnant women, may translate to better pregnancy outcomes. HIV testing is a marker for a more thorough general work-up. However, further studies are required in order to strengthen our conclusions.

## Figures and Tables

**Table 1 tropicalmed-07-00242-t001:** Simplified recommended testing scheme for pregnant women in Poland (According to Ordinance of the Minister of Health on the Standard of Organizational Perinatal Care).

Examination Date	Diseases that Pregnant Women Ought to Be Tested for
Up to the 10th week of gestation or at the time of first reporting	1. syphilis 2. HIV 3. HCV 4. Toxoplasmosis 5. Rubella
Week 21–26th of gestation	In women with negative results in the first trimester—testing for toxoplasmosis.
Week 33–37th of gestation	1. HBV 2. HIV 3. Vaginal and rectal culture for B-hemolytic streptococci (35–37 weeks of gestation). 4. syphilis and HCV testing in a group of women with an increased risk of infection.

**Table 2 tropicalmed-07-00242-t002:** Baseline characteristics and clinical data on women with suspected primary *Toxoplasma gondii* infection with both correctly and incorrectly implemented recommended testing for human immunodeficiency virus (HIV) infection in pregnant women in Poland.

Characteristic	Total n = 273	Correctly Implemented Testing towards HIV n = 243	Incorrectly Implemented Testing towards HIV n = 30	*p*-Value
Age in years, median [IQR]	30 [26–33]	30 [26–33]	29 [26.25–34.5]	0.8337
Living in rural area, n (%)	119 (43.6)	103 (42.4)	16 (53.3)	0.2540
History of miscarriage, n (%)	44 (16.1)	37 (15.2)	7 (23.3)	0.2546
Good socioeconomic status, n (%)	267 (97.8)	238 (97.9)	29 (96.7)	0.6530
Chronic diseases, n (%)	69 (25.3)	65 (26.7)	4 (13.3)	0.1243
Autoimmune diseases, n (%)	53 (19.4)	50 (20.6)	3 (10)	0.2231
Clinical Evaluation				
Confirmed primary toxoplasmosis	74 (27.1)	63 (25.9)	11 (36.7)	0.2118
Toxoplasmosis in the past	114 (41.8)	103 (42.4)	11 (36.7)	0.5489
Inconclusive results	14 (5.1)	13 (5.3)	1 (3.3)	1.0000
Excluded toxoplasmosis	71 (26)	64 (26.3)	7 (23.3)	0.7234
Correct ultrasound result, n (%)	251 (91.9)	234 (96.3)	11 (36.7)	<0.00001
Correctly implemented all recommended testing	172 (63)	171 (70.4)	1 (3.3)	<0.00001
Lymphadenopathy, n (%)	14 (5.1)	14 (5.7)	0 (0)	0.3772
Influenza-like symptoms, n (%)	35 (12.8)	31 (12.8)	4 (13.3)	1.0000
Both lymphadenopathy and influenza-like syndrome, n (%)	41 (15)	37 (15.2)	4 (13.3)	1.0000

**Table 3 tropicalmed-07-00242-t003:** Univariate logistic regression analysis of factors associated with correct implementation of testing for HIV infection during pregnancy.

	Univariate		
Factor	Odds Ratio	95% Confidence Interval	*p*-Value
Age in years, median [IQR]	1.0	0.9–1.1	0.88000
Living in rural area, n (%)	1.2	0.5–2.7	0.62000
History of miscarriage, n (%)	0.5	0.2–1.3	0.18000
Good socioeconomic status, n (%)	1.6	0.2–12.1	0.65000
Chronic diseases, n (%)	3.1	0.9–10.5	0.07000
Confirmed primary toxoplasmosis	1.1	0.5–2.8	0.79000
Toxoplasmosis in the past	0.8	0.4–1.8	0.59000
Inconclusive results	1.5	0.2–12.0	0.69000
Excluded toxoplasmosis	1.1	0.4–2.6	0.89000
Correct ultrasound result, n (%)	44.9	16.6–121.8	<0.00001
Correctly implemented all recommended testing	68.9	9.2–515.3	<0.00001
Lymphadenopathy, n (%)	1.0	1.0–1.0	0.99000
Influenza-like symptoms, n (%)	1.2	0.3–4.2	0.76000
Both lymphadenopathy and influenza-like syndrome, n (%)	1.5	0.4–5.3	0.76000

## Data Availability

The data sets used and/or analyzed during the current study can be made available by the corresponding author on reasonable request.

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
