# Peer review of "Testing for HIV Increases the Odds of Correct Fetal Ultrasound Result"

_tropicalmed, 2022, doi:10.3390/tropicalmed7090242_

Round 1

Reviewer 1 Report

Comments to authors

This is a well-written paper on a study that explored associations between HIV testing in pregnancy and other related health outcomes. Below, please find my questions and suggestions for improvement.

Overall: There are some minor typos and grammatical errors in the manuscript.

Title: The title is misleading, unless you are arguing that correct fetal ultrasounds are appropriate clinical follow up of pregnant women? Perhaps the title should be “Testing for HIV Increases the Odds of Correct Fetal Ultrasound Result” which is your main finding, correct?

Abstract

The abstract is thorough and well written. However, it is well over the suggested word count of 200 words. I will leave it up to the Editors to determine if that is acceptable. Also, the interquartile range of your sample in the abstract does not match with the interquartile range in the results section. I suggest double checking that and all other numbers to ensure accuracy and consistency of reporting.

Introduction

I suggest deleting the word, “the utilization of” in the purpose statement. It doesn’t seem that you were assessing if people choose to have HIV tests, but rather simply assessing HIV testing among pregnant women. Similarly, I would delete the word, “inappropriate” in the final sentence of the introduction. It seems you were assessing patterns associated with HIV testing whether they were appropriate or not.

Including some brief information about the incidence and prevalence of HIV in Poland would be useful to include.

Methods

Please provide a rationale for why only women suspected of Toxoplasma gondii were included in the study in the methods section. Then, in the limitations section, please explore how these women might be different than other pregnant women not suspected of the infection and how including only these women may have influenced results.

Also, please provide definitions for your outcomes of interest and/or a description of how they are collected. For instance, much of the paper focuses on correct fetal ultrasound result, but we don’t know what that means. Is it that a technician correctly completed the ultrasound? That the baby was healthy? Or, perhaps both?

Is anamnesis the correct word in this situation? And if you list all of the variables that were collected here, the list should include all of the demographic variables collected. For example, place of residence (rural etc.) comorbidities etc. were not included.

Results

As previously mentioned, the interquartile range of the sample is not consistent between the abstract and results does not match. Please review all results to ensure accuracy and consistent reporting.

Discussion

Because not testing for HIV was so closely associated with not testing for other infections, it seems likely that providers just weren’t ordering the tests, not that women were opting out of them. Also, since these tests are mandated by law, isn’t it the providers’ responsibility to order the tests, not womens’ responsibility to request them?

The authors should strengthen their rationale regarding the need for HIV testing among pregnant women if there are not HIV infections appearing with testing. It seems unlikely that just by getting an HIV test, you are more likely to have better fetal ultrasounds. So a clearer link between these two situations is needed. As the article is written now, the conclusion that HIV testing can lead to better health outcomes is not justified.

Also, exploring reasons why 11% of women don’t get tested would be interesting to include. This is also an area for future study.

Author Response

Reviewer 1

Point 1:

This is a well-written paper on a study that explored associations between HIV testing in pregnancy and other related health outcomes. Below, please find my questions and suggestions for improvement.

 Overall: There are some minor typos and grammatical errors in the manuscript.

Response: Thank you for the acknowledgment of our work.

Point 2: Title: The title is misleading, unless you are arguing that correct fetal ultrasounds are appropriate clinical follow up of pregnant women? Perhaps the title should be “Testing for HIV Increases the Odds of Correct Fetal Ultrasound Result” which is your main finding, correct?

Response: Thank you for this remark, we have implemented it.

Point 3: Abstract

The abstract is thorough and well written. However, it is well over the suggested word count of 200 words. I will leave it up to the Editors to determine if that is acceptable. Also, the interquartile range of your sample in the abstract does not match with the interquartile range in the results section. I suggest double checking that and all other numbers to ensure accuracy and consistency of reporting.

Response: Thank you for acknowledgment of our work. In the original version of the abstract we have provided the age range instead of age interquartile range, therefore, we have corrected the mistake.

Point 4: Introduction

I suggest deleting the word, “the utilization of” in the purpose statement. It doesn’t seem that you were assessing if people choose to have HIV tests, but rather simply assessing HIV testing among pregnant women. Similarly, I would delete the word, “inappropriate” in the final sentence of the introduction. It seems you were assessing patterns associated with HIV testing whether they were appropriate or not.

Response: Thank you for these remarks. We have implemented them.

Point 5: Including some brief information about the incidence and prevalence of HIV in Poland would be useful to include.

Response: Thank you for this remark. We have included these information in the introduction.

Point 6: Methods

Please provide a rationale for why only women suspected of Toxoplasma gondii were included in the study in the methods section. Then, in the limitations section, please explore how these women might be different than other pregnant women not suspected of the infection and how including only these women may have influenced results.

Response: Thank you for this remark. In our Hospital we consult pregnant women with suspicion of toxoplasmosis and/or other infectious diseases (rarely). Therefore in order to unify the cohort we decided to include only women with suspected toxoplasmosis. We have included this information in the methods section. Moreover, we have explained in the limitations section how these women might be different than other pregnant women.

Point 7: Also, please provide definitions for your outcomes of interest and/or a description of how they are collected. For instance, much of the paper focuses on correct fetal ultrasound result, but we don’t know what that means. Is it that a technician correctly completed the ultrasound? That the baby was healthy? Or, perhaps both?

Response: Thank you for this remark. We have added the outcome definition in the methods section.

Point 8: Is anamnesis the correct word in this situation? And if you list all of the variables that were collected here, the list should include all of the demographic variables collected. For example, place of residence (rural etc.) comorbidities etc. were not included.

Response: Thank you for this remark. We have corrected it.

Point 9: Results

As previously mentioned, the interquartile range of the sample is not consistent between the abstract and results does not match. Please review all results to ensure accuracy and consistent reporting.

Response: Thank you for this remark. We have corrected the mistake.

Point 10: Discussion

Because not testing for HIV was so closely associated with not testing for other infections, it seems likely that providers just weren’t ordering the tests, not that women were opting out of them. Also, since these tests are mandated by law, isn’t it the providers’ responsibility to order the tests, not womens’ responsibility to request them?

Response: Thank you for this question. Yes, it is the provider’s responsibility to order the tests. However, despite the law 11% of our cohort had not been tested for HIV during their 1st trimester, but only in our Hospital (where they should be consulted with already one HIV test performed).

Point 11: The authors should strengthen their rationale regarding the need for HIV testing among pregnant women if there are not HIV infections appearing with testing. It seems unlikely that just by getting an HIV test, you are more likely to have better fetal ultrasounds. So a clearer link between these two situations is needed. As the article is written now, the conclusion that HIV testing can lead to better health outcomes is not justified.

Response: Thank you for this remark. Correct fetal ultrasound results were more likely to be obtained in women tested for HIV according to recommendations. This suggests that holistic approach in screening, both for communicable and non-communicable diseases, among pregnant women may translate to better pregnancy outcomes. The conclusion is based on the logistic regression results. However, we agree that further studies are required in this matter to strengthen these conclusions.

Point 12: Also, exploring reasons why 11% of women don’t get tested would be interesting to include. This is also an area for future study.

Response: Thank you for this remark. We agree with the reviewer that reasons why these 11% of women had not been tested for HIV are interesting to include. However, we do not have these reasons. We can only guess that for some women the reason for not getting tested might be the suspicion of infidelity, stigma related to the HIV, scare. However, without having these reasons confirmed we couldn’t provide in our manuscript our guesses for reasons why women had not been tested for HIV.

Reviewer 2 Report

This is a worthwhile study which was unfortunately disrupted by the COVID-19 pandemic. I think the abstract should make clear that this study was done at an Infectious Disease Hospital since that sheds light on the fact that an 89% HIV testing rate is surprisingly low. I also think that one might conclude that HIV testing is a marker for a more thorough general work-up. I believe lines 88 to 95 could be condensed to simply list the variables that showed no significant differences rather than use the term "more likely to..." and then go on to list p values that show no significant differences.

Author Response

Reviewer 2

Point 1: This is a worthwhile study which was unfortunately disrupted by the COVID-19 pandemic. I think the abstract should make clear that this study was done at an Infectious Disease Hospital since that sheds light on the fact that an 89% HIV testing rate is surprisingly low.

Response: Thank you for this remark. We have added that information. These women when being consulted in our hospital had not been tested for HIV according to recommendations (1st test performed during 1st trimester). However, in our Hospital all of these women were tested and none of them had the HIV-infection confirmed.

Point 2: I also think that one might conclude that HIV testing is a marker for a more thorough general work-up.

Response: Thank you for this remark. We have added this conclusion.

Point 3: I believe lines 88 to 95 could be condensed to simply list the variables that showed no significant differences rather than use the term "more likely to..." and then go on to list p values that show no significant differences.

Response: Thank you for this remark. We have implemented it.

Reviewer 3 Report

This manuscript mainly investigated how HIV Testing might affect the follow up medical treatment of pregnant women in Poland. Basically, there are numerous major and minor concerns regarding this manuscript.

1. The major conclusion of this study is that the screening of HIV testing among pregnant women may translate to better pregnancy outcomes. As a retrospective analysis of data collected from medical records, it cannot prove causality, just an association.

2. Selection bias may exist because the participants were recruited from only one hospital. How representative is the sample collected through only one site with a small number? Can the conclusions of this manuscript be extrapolated to Poland as a whole?

3. This study investigated only 273 pregnant women. Please state how to calculate the minimum sample size of participants for this survey.

4. Lack of the inclusion and exclusion criteria.

5. All Tables should be listed in three-line table.

Author Response

Reviewer 3

Point 1: This manuscript mainly investigated how HIV Testing might affect the follow up medical treatment of pregnant women in Poland. Basically, there are numerous major and minor concerns regarding this manuscript.

Response: Thank you for this remark and for all tips on how to improve our manuscript.

Point 2: The major conclusion of this study is that the screening of HIV testing among pregnant women may translate to better pregnancy outcomes. As a retrospective analysis of data collected from medical records, it cannot prove causality, just an association.

Response: Thank you for this remark. We agree with the reviewer. We have corrected the conclusions.

Point 3: Selection bias may exist because the participants were recruited from only one hospital. How representative is the sample collected through only one site with a small number? Can the conclusions of this manuscript be extrapolated to Poland as a whole?

Response: Thank you for this remark. Our study was conducted in the biggest Infectious Diseases Hospital in Poland. Therefore, our results may be extrapolated to Poland as a whole. We believe our conclusions may be a start for further discussion in this matter. However, further studies are required in order to strengthen our conclusions. Therefore, we have added this statement in our conclusions.

Point 4: This study investigated only 273 pregnant women. Please state how to calculate the minimum sample size of participants for this survey.

Response: Thank you for this question. We did not calculate the sample size for this survey. We have included all women consulted in our Hospital since consultations of pregnant women with suspected toxoplasmosis started in our Hospital. However, our cohort is only 273 due to transforming our Hospital into a COVID-19 – only facility. Therefore, further consultations could not occur.

Point 5: Lack of the inclusion and exclusion criteria.

Response: Thank you for this remark. We have stated the inclusion and exclusion criteria in the methods section: We performed a retrospective analysis of data collected from medical records of all consecutive pregnant women, referred by their gynecologists to the Hospital for Infectious Diseases in Warsaw due to suspected primary Toxoplasma gondii (TG) infection, between 1st September 2019 and 14th March 2020. In our Hospital we consult pregnant women with suspicion of toxoplasmosis and/or other infectious diseases (rarely). Therefore, in order to unify the cohort we decided to include only women with suspected toxoplasmosis.

Point 6: All Tables should be listed in three-line table.

Response: Thank you for this remark. All Tables ale cited in the manuscript.

Round 2

Reviewer 3 Report

As I suggested before, "The major conclusion of this study is that the screening of HIV testing among pregnant women may translate to better pregnancy outcomes. As a retrospective analysis of data collected from medical records, it cannot prove causality, just an association." However, there is still a  causality-directed title "Testing for HIV Increases the Odds of Correct Fetal Ultrasound Result" and the corresponding abstract and main-text.

Authors fail to revise the question "All Tables should be listed in three-line table."